# Association between Body Composition, Physical Activity Profile, and Occurrence of Knee and Foot Postural Alterations among Young Healthy Adults

**Sadaf Ashraf [1], Roberto Viveiros [1], Cíntia França [1,2,3,*], Rui Trindade Ornelas [1] and Ana Rodrigues [1,3]**

1   Department of Physical Education and Sport, University of Madeira, 9020-105 Funchal, Portugal; 2136420@student.uma.pt (S.A.); roberto.viveiros@staff.uma.pt (R.V.); rui.ornelas@staff.uma.pt (R.T.O.); anajar@staff.uma.pt (A.R.)
2   Laboratory for Robotic and Engineering Systems, Interactive Technologies Institute, 9020-105 Funchal, Portugal
3   Research Center in Sports Sciences, Health Sciences and Human Development (CIDESD), 5000-801 Vila Real, Portugal
*   Correspondence: cintia.franca@staff.uma.pt

**Abstract:** Knee and foot deformities refer to structural abnormalities in the knee and foot bones, joints, ligaments, or muscles. Various factors, including genetics, injury, disease, or excessive use, can cause these deformities. These musculoskeletal conditions can significantly impact individuals' quality of life. This study examined foot and knee deformities in 231 young healthy adults (165 men, 66 women) aged 22.6 ± 4.9 years and their association with physical activity and body composition. The postural assessment was performed by two Physiotherapists, with the subject standing in three views: side, anterior, and posterior. Physical activity (Baecke's Habitual Physical Activity Questionnaire) and body composition (InBody 770) were assessed. Results showed that the most common foot deformity was *pes planus*, while the *genu recurvatum* was the most common knee deformity among the individuals. Physical activity level was negatively associated with knee and foot deformities. Conversely, body composition differed with the presence of *genu recurvatum*. These findings present a starting point to understand the occurrence of knee and foot postural alterations according to the individuals' body composition and physical activity profiles, which could support the deployment of tailored interventions among healthy adults. In addition, early detection of postural changes is crucial in mitigating their negative long-term impact on physical well-being.

**Keywords:** posture; *genu recurvatum*; *pes planus*; *pes cavus*

## 1. Introduction

Posture is an attitude or body position maintained at rest, adopted in an activity, or an exact way of supporting the body. It makes quick modifications over time to reduce the body's energy, leading to a rise in muscle stress to maintain proper balance [1]. Posture is maintained by skeletal muscle contraction, coordinated by a series of external stimuli, and through continuous neuromuscular type modifications, representing the body's reaction to the force of gravity [2]. Each individual's distinctive posture may be influenced by several factors, such as ligament elasticity, breathing issues, or bone malformations [3]. Thus, postural changes have been related to unbalanced body components, leading to increased muscle stress to maintain proper balance [4], resulting in discomfort and functional impairment [5].

Studies emphasize that one of the most urgent issues facing contemporary society is the widespread nature of postural abnormalities [6–9], which might eventually lead to inevitable injuries and pain in the corresponding body organ. Foot and knee postures are known to determine how well the lower extremity operates and may contribute to the individual's propensity for recurrent trauma [10,11]. For instance, a study conducted to assess

the prevalence of foot pain and deformity among the Danish population reported high correlations between foot pain and foot alterations [12]. Additionally, the previous literature has emphasized the relationship between knee and foot misalignment, with changes in foot alignment before and after knee surgery [13]. Foot postures like pronated foot and flat foot (*pes planus*) are significantly associated with changes in dynamic function, alignment, and medial compartment knee osteoarthritis [14]. Several elements, including lifestyle choices, employment, and the environment, endanger the skeletal system [15]. According to previous studies, decreased physical activity (PA) in overweight and obese individuals contributes to altered lower-limb joint loading compared to normal-weight adults [16]. Differences in walking patterns between overweight and normal-weight individuals are associated with the attempt to increase stability due to impaired balance, minimize external work, and decrease load at the knee [17]. On the other hand, foot problems might be derived from the effects of increased weight on plantar loading, underlining a link between obesity and the development of foot pain [18].

Previous research has identified the most common knee and foot deformities, such as *genu recurvatum*, *genu varum*, *genu varus*, *pes planus*, and *pes cavus*. *Genu Recurvatum* is the position of a tibiofemoral joint in which the range of motion occurs beyond neutral or 0° of extension [19]. In *genu valgus* deformity, ankle joints are spaced apart when knees are in contact with each other in a weight-bearing position [20]. In contrast, in *genu varus*, the internal condyles of the femur become spaced apart if they are in weight-bearing contact with the media, malleolus of the ankle [20]. The medial longitudinal arch is clinically significant in diseases and the functioning of joints and muscles of the ankle and knee [21]. When the foot's medial arch is disturbed, deformities occur, whereby considerable reduction in the arch causes *pes planus*, whereas the increase in the arch exceeding 18 mm causes *pes cavus* [22].

Meanwhile, the literature has reported that an increase in body mass index (BMI) leads to postural instability in young adults who are obese and non-obese [23–25]. A study on youngsters aged between 7 and 16 years revealed a substantial correlation between age, BMI, and *genu varum* [26]. Weakness in the lower extremities can lead to challenges in everyday activities and regular movements. Enhancing PA levels might be crucial to fighting overweight and obesity, diminishing the risk of injury, and improving the quality of life [27–29]. However, details regarding the relationship between PA, body composition variables, and the prevalence of foot and knee deformities are still lacking among healthy adults. Providing an early diagnosis of postural impairment allows designing targeted interventions that may reduce the detrimental effects of postural deformities [30]. It helps physical education teachers, coaches, and healthcare professionals design appropriate and corrective exercises to improve body alignment. Therefore, the present study aims to (1) determine the occurrence of postural alterations in the knee and foot among healthy adults and (2) compare the PA and body composition profiles between individuals with and without postural alterations in the knee and foot.

## 2. Methodology

### 2.1. Sample

This study included 231 adults who were students from the Physical Education and Sports course at the University of Madeira. The sample included 165 men and 66 women aged between 18 and 44 years old (22.5 ± 4.2 years). All participants were healthy and not injured at the time of data collection. The inclusion and exclusion criteria are presented in Figure 1. The optimal sample size was calculated using G*Power [31]. A priori independent samples *t*-test indicated a total sample of 128 participants (64 in each group) to attain 80% power for an effect size of 0.50 at the 0.05 level of significance. All the procedures implemented in the current study received ethical approval from the Scientific Committee of The Faculty of Physical Education and Sports at the University of Madeira (reference: ACTA N.77-12 April 2016). All participants were volunteers, and informed consent was signed before data collection.

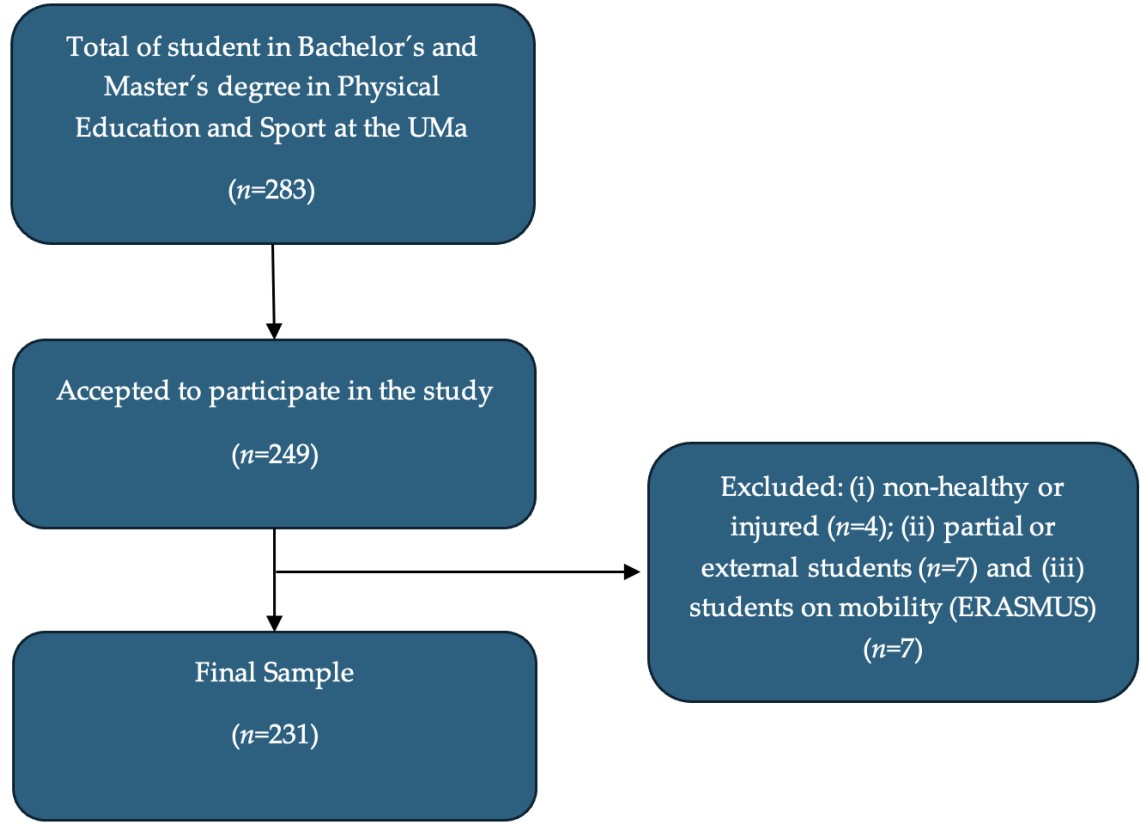

**Figure 1.** Participants' inclusion and exclusion criteria (*n* = 231).

*2.2. General Procedure*

2.2.1. Postural Evaluation

The postural evaluation was conducted using the Postural Assessment Table, employing the visual assessment method. The evaluation was carried out by two experienced physiotherapists who were trained in postural assessment techniques. A similar process of postural assessment was employed, as described by [32], in a study involving rhythmic gymnasts and non-trainees to observe the presence of postural alterations. The subjects under observation were assessed while standing in three distinct views: anterior, side, and posterior. Before the assessment, participants were instructed to present themselves barefoot and in light clothing. They assumed a neutral standing position, with specific guidance given regarding their gaze direction (looking forward and fixing their gaze on a point on a wall). Data collection spanned two weeks, during which assessments were conducted for each relevant class. These assessments took place in a dedicated physical performance laboratory between 10 a.m. and 2 p.m. on weekdays. Each evaluation session lasted approximately 40 min, allowing sufficient time for participants to complete consent forms and undergo testing. The decision to conduct assessments during regular working hours was deliberate, aiming to mitigate potential biases related to participants' alertness and concentration levels. Moreover, this approach ensured consistent conditions for all participants, promoting the overall validity of the research findings. By allocating uniform time slots for assessment sessions, the study sought to enhance the reliability and credibility of the collected data.

(a)    In the Anterior View

Alignment and symmetry of knee and foot were observed, including alignment of the knee (internal, external rotation, varus, and valgus) and malleolus (symmetry and asymmetry).

(b)    In the Side View

Alignment and symmetry of knee and foot were observed, including Knees (*genu recurvatum*) and feet (medial longitudinal arch for *pes planus* and *pes cavus*).

(c)    In the Posterior View

Alignment and symmetry of knee and foot were observed, including popliteal lines (symmetry and asymmetry) and feet (valgus and varus).

### 2.2.2. Body Composition

Stature was measured to the nearest 0.01 cm using a stadiometer (SECA 213, Hamburg, Germany). Body composition variables were assessed using hand-to-foot bioelectrical impedance analysis (InBody 770, Cerritos, CA, USA). Body mass, body mass index (BMI), fat mass percentage (FM%), total body water (TBW), Intracellular Water, Extracellular Water, protein, minerals, and skeletal muscle mass (SMM), and waist–hip ratio WHR (cm).

### 2.2.3. Physical Activity

Baecke's Habitual PA questionnaire [33] was used to calculate PA, which includes the following variables:

(a)    Formal PA

This variable assesses an individual's engagement in formal PA and structured sports. Participants report the activity type, each associated with an intensity factor (0.76 for light, 1.26 for moderate, and 1.76 for vigorous). Time spent per week (0.5 to 4.5 h) and proportion of the year (0.04 to 0.92) are considered. Scores are calculated by multiplying intensity, time, and proportion. The final score, within predefined ranges, reflects overall activity.

(b)    Informal PA

This variable assesses informal activities in daily life unrelated to structured exercise. Responses to Likert scale questions gauge frequency and intensity. Higher scores indicate greater activity levels.

(c)    Total Practical History

Scores from Formal PA and Informal PA are combined to measure overall habitual PA. This encompasses both structured and informal activities.

(d)    Frequency of PA

Assesses planned exercise sessions beyond daily activities. The score typically reflects reported exercise sessions per week.

### *2.3. Statistical Analysis*

Data exploration was carried out to identify possible data entry errors and the presence of outliers. Descriptive statistics were presented as means ± standard deviation. All data were checked for normality using the Kolmogorov–Smirnov test. The independent samples *t*-test was conducted to examine the differences between participants with and without postural changes in quantitative variables with a normal distribution. The software used was SPSS version 27.0 (SPSS Inc., Chicago, IL, USA), and the significance level adopted was 5%.

## 3. Results

Figure 2 summarizes the occurrence of postural alterations of the knee and foot among the participants based on gender. Overall, *genu recurvatum* was the most frequent postural alteration, representing 54.1%. In contrast, *pes cavus* (4.3%) and *genu varus* (8.7%) were the least frequent postural alterations. A higher proportion of females presented *genu recurvatum*, while *knee valgus* was more prevalent among men. At the same time, the opposite occurs in *knee valgus* (60.4% vs. 39.6%), with the diagnosis occurring in a greater proportion in males than females.

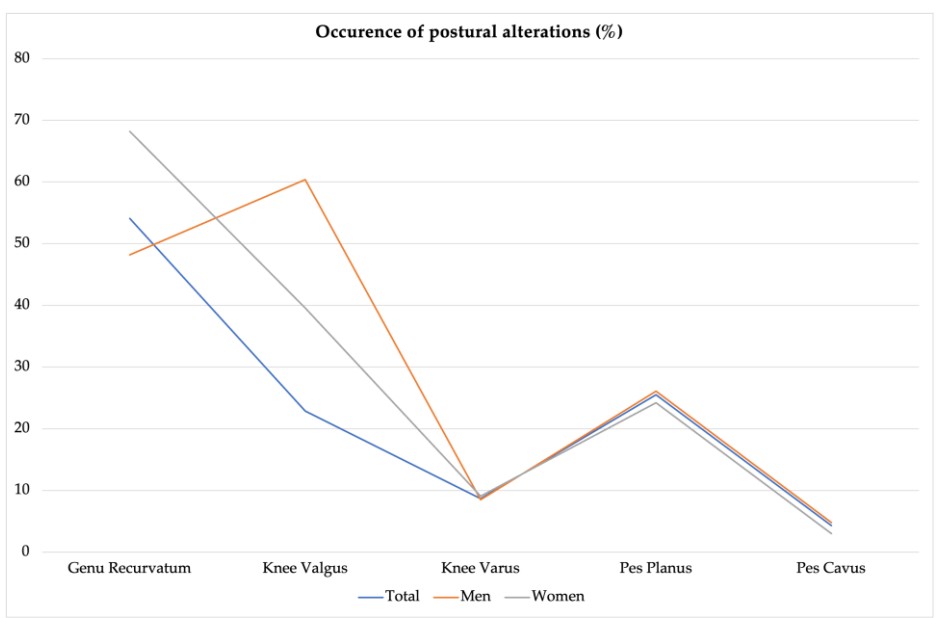

**Figure 2.** Occurrence of knee and foot postural alterations among the participants.

Table 1 presents the descriptive statistics for body composition variables according to knee postural changes and the comparison results between groups (with and without knee postural changes). Although without statistically significant differences, body mass (M = 70.7 ± 11.7 kg) and BMI (M = 23.4 ± 2.6 kg/m$^2$) in individuals without *genu recurvatum* alteration were higher than body mass (M = 68.6 ± 12.2 kg) and BMI (M = 23.3 ± 3.2 kg/m$^2$) in individuals with *genu recurvatum* alteration. Furthermore, WHR (M = 0.86 ± 0.07 cm) and FM (M = 20.8 ± 10.1%) in individuals with *genu recurvatum* alteration were significantly higher than WHR (M = 0.82 ± 0.04 cm) and FM (M = 14.9 ± 7.0%) in individuals without *genu recurvatum* alteration. Stature, TBW, Intracellular Water, Extracellular Water, protein, minerals, and SMM in individuals without *genu recurvatum* alteration were significantly higher in individuals with an alteration ($p < 0.05$).

Body composition, including stature, body mass, TBW, intracellular and extracellular water, protein, FM%, and SMM in individuals without *genu valgus* and *genu varus* was higher than in individuals with *genu valgus* and *genu varus*. The difference was not statistically significant.

Overall, average body mass and BMI were not statistically significant, with a prevalence of *genu recurvatum*. At the same time, other body composition parameters were statistically significant with the occurrence of *genu recurvatum* ($p < 0.05$). Furthermore, Body composition showed no statistically significant relation with *knee valgus* and *varus* prevalence.

Table 2 resumes the descriptive statistics for body composition variables according to foot postural changes and the comparison results between groups (with and without postural changes). Overall, no statistically significant differences were observed between groups regarding body composition on the *pes planus* deformities. The same trend was seen concerning *pes cavus*, except for stature, which was significantly larger among individuals who showed *pes cavus* postural changes ($p = 0.035$).

**Table 1.** Descriptive statistics for body composition variables according to knee postural changes and the comparison results between groups (with and without knee postural changes).

| Variable | *Genu Recurvatum* | | | | *Knee Valgus* | | | | *Knee Varus* | | | |
| | With | Without | Comparison | | With | Without | Comparison | | With | Without | Comparison | |
| | Mean ± SD | Mean ± SD | t | p | Mean ± SD | Mean ± SD | t | p | Mean ± SD | Mean ± SD | t | p |
|---|---|---|---|---|---|---|---|---|---|---|---|---|
| Stature (cm) | 171.21 ± 9.79 | 174.31 ± 8.44 | 2.497 | 0.013 * | 171.35 ± 9.97 | 172.93 ± 9.12 | 1.052 | 0.294 | 172.57± 9.50 | 172.95 ± 7.56 | −0.167 | 0.867 |
| Body Mass (kg) | 68.64 ± 12.23 | 70.68 ± 11.72 | 1.222 | 0.218 | 68.89 ± 11.53 | 69.72 ± 12.21 | 0.679 | 0.670 | 67.43 ± 7.84 | 69.64 ± 12.31 | 0.168 | 0.279 |
| BMI (kg/m$^2$) | 23.27 ± 3.15 | 23.35 ± 2.64 | −0.293 | 0.769 | 23.43 ± 3.40 | 23.41 ± 2.79 | −0.030 | 0.976 | 2340 ± 2.91 | 23.09 ± 2.16 | 0.468 | 0.640 |
| TBW (L) | 40.48 ± 8.37 | 45.18 ± 7.67 | 2.897 | 0.005 ** | 40.21 ± 8.43 | 43.76 ± 8.14 | 1.907 | 0.060 | 42.71 ± 8.34 | 45.07 ± 9.47 | −0.481 | 0.632 |
| Intracellular Water (L) | 25.56 ± 5.41 | 28.61 ± 4.96 | 2.909 | 0.005 ** | 25.39 ± 5.43 | 27.69 ± 5.28 | 1.911 | 0.059 | 27.01 ± 5.40 | 28.60 ± 6.16 | −0.491 | 0.624 |
| Extracellular Water (L) | 14.92 ± 2.91 | 16.57 ± 2.72 | 2.863 | 0.005 ** | 14.82 ± 3.01 | 16.07 ± 2.87 | 1.892 | 0.062 | 16.50 ± 3.31 | 15.71 ± 2.96 | −0.459 | 0.647 |
| Protein (kg) | 11.05 ± 2.37 | 12.37 ± 2.14 | 2.911 | 0.004 ** | 10.97 ± 2.32 | 11.97 ± 2.29 | 1.909 | 0.059 | 11.67 ± 2.33 | 12.30 ± 2.64 | −0.457 | 0.649 |
| Minerals (kg) | 3.92 ± 0.80 | 4.34 ± 0.76 | 2.704 | 0.008 ** | 4.22 ± 0.79 | 3.87 ± 0.79 | 1.984 | 0.050 * | 4.48 ± 0.99 | 4.12 ± 0.80 | −0.773 | 0.441 |
| FM (%) | 20.78 ± 10.12 | 14.92 ± 7.01 | −3.333 | 0.001 ** | 17.05 ± 8.46 | 20.16 ± 10.75 | −1.502 | 0.136 | 17.77 ± 5.95 | 17.91 ± 9.31 | 0.027 | 0.979 |
| SMM (kg) | 31.32 ± 7.05 | 35.32 ± 6.47 | 2.918 | 0.004 ** | 31.09 ± 7.07 | 34.11 ± 6.89 | 1.921 | 0.058 | 33.22 ± 7.04 | 35.27 ± 8.01 | −0.495 | 0.622 |
| WHR (cm) | 0.86 ± 0.07 | 0.82 ± 0.04 | −3.045 | 0.003 ** | 0.85 ± 0.06 | 0.84 ± 0.06 | −0.665 | 0.507 | 0.85 ± 0.06 | 0.84 ± 0.06 | −0.213 | 0.832 |

BMI (body mass index), TBW (total body water), FM (fat mass), SMM (skeletal muscle mass), WHR (waist–hip ratio), SD (standard deviation), * $p \leq 0.05$, ** $p \leq 0.01$.

**Table 2.** Descriptive statistics for body composition variables according to foot postural changes and the comparison results between groups (with and without postural changes).

| Variable | Pes Planus | | | | Pes Cavus | | | |
|---|---|---|---|---|---|---|---|---|
| | With | Without | Comparison | | With | Without | Comparison | |
| | Mean ± SD | Mean ± SD | t | p | Mean ± SD | Mean ± SD | t | p |
| Stature (cm) | 171.47 ± 9.20 | 172.94 ± 9.39 | 1.019 | 0.309 | 176.50 ± 5.10 | 172.36 ± 9.46 | −2.372 | 0.035 * |
| Body Mass (kg) | 69.56 ± 12.27 | 69.51 ±11.98 | −0.022 | 0.982 | 72.37 ± 6.55 | 69.38 ± 12.23 | −0.765 | 0.445 |
| BMI (kg/m$^2$) | 23.96 ± 3.31 | 23.22 ±2.76 | 0.093 | 0.108 | 23.35 ± 1.77 | 23.42 ± 2.96 | −1.613 | 0.945 |
| TBW (L) | 42.99 ± 8.27 | 42.99 ± 8.27 | 0.479 | 0.633 | 42.65 ± 8.36 | 45.83 ± 8.08 | 0.479 | 0.458 |
| Intracellular Water (L) | 26.54 ± 5.58 | 27.19 ± 5.37 | 0.488 | 0.627 | 26.96 ± 5.41 | 29.05 ± 5.22 | 0.488 | 0.453 |
| Extracellular Water (L) | 15.56 ± 3.13 | 15.80 ± 2.91 | 0.460 | 0.647 | 15.68 ± 2.96 | 16.78 ± 2.87 | 0.460 | 0.471 |
| Protein (kg) | 11.76 ± 2.32 | 11.46 ± 2.39 | 0.511 | 0.611 | 11.65 ± 2.34 | 12.53 ± 2.25 | 0.511 | 0.469 |
| Minerals (kg) | 4.15 ± 0.801 | 4.03 ± 0.82 | 0.587 | 0.558 | 4.11 ± 0.80 | 4.54 ± 0.84 | 0.587 | 0.301 |
| FM (%) | 17.53 ± 8.61 | 19.29 ± 11.25 | −0.776 | 0.439 | 17.98 ± 9.37 | 16.25 ± 3.49 | −0.776 | 0.715 |
| SMM (kg) | 33.46 ± 7.01 | 32.61 ± 7.27 | 0.590 | 0.627 | 33.17 ± 7.06 | 35.90 ± 6.81 | 0.487 | 0.450 |
| WHR (cm) | 0.85 ± 0.06 | 0.84 ± 0.06 | −0.423 | 0.673 | 0.84 ± 0.03 | 0.84 ± 0.06 | −0.423 | 0.863 |

BMI (body mass index), FM (fat mass), TBW (total body water), WHR (waist–hip ratio), SMM (skeletal muscle mass), SD (standard deviation), * $p \leq 0.05$.

Table 3 displays the descriptive statistics for PA variables according to knee postural changes and the comparison results between groups (with and without postural changes). The results indicate a significantly larger Score of Formal PA among individuals who presented *knee valgus* compared to those unaffected by this deformity ($p = 0.048$). In addition, a substantially higher Score of Formal PA ($p = 0.003$) and Total Score ($p = 0.004$) was identified in the group with *knee varus* compared to the individuals that did not present this postural change.

**Table 3.** Descriptive statistics for PA variables according to knee postural changes and the comparison results between groups (with and without postural changes).

| Variable | *Genu Recurvatum* | | | | *Knee Valgus* | | | | *Knee Varus* | | | |
| | With | Without | Comparison | | With | Without | Comparison | | With | Without | Comparison | |
| | Mean ± SD | Mean ± SD | t | p | Mean ± SD | Mean ± SD | t | p | Mean ± SD | Mean ± SD | t | p |
| Score Formal PA | 11.93 ± 2.91 | 12.06 ± 3.10 | 0.292 | 0.770 | 12.70 ± 3.11 | 11.72 ± 2.91 | −1.989 | 0.048 * | 13.62 ± 1.66 | 11.85 ± 3.03 | −3.427 | 0.003 ** |
| Score Informal PA | 2.78 ± 0.58 | 2.81 ± 0.52 | 0.283 | 0.778 | 2.76 ± 0.57 | 2.81 ± 0.55 | 0.582 | 0.561 | 2.78 ± 0.67 | 2.79 ± 0.54 | 0.132 | 0.895 |
| Score Total | 14.89 ± 3.28 | 14.73 ± 3.10 | 0.339 | 0.735 | 15.46 ± 3.22 | 14.54 ± 3.13 | −1.730 | 0.085 | 16.57 ± 1.93 | 14.64 ± 3.21 | −3.279 | 0.004 ** |
| Practical History | 9.90 ± 5.64 | 9.34 ± 5.44 | −0.656 | 0.513 | 10.39 ± 5.11 | 9.38 ± 5.69 | −1.054 | 0.293 | 9.13 ± 4.68 | 9.75 ± 5.66 | 0.457 | 0.648 |
| Frequency of PA | 8.42 ± 4.57 | 7.91 ± 3.61 | −0.696 | 0.488 | 8.89 ± 5.10 | 7.94 ± 3.79 | −1.152 | 0.251 | 8.77 ± 4.66 | 8.16 ± 4.18 | −0.494 | 0.622 |

PA (physical activity); * $p \leq 0.05$, ** $p \leq 0.01$.

Finally, Table 4 resumes the descriptive statistics for PA variables according to the foot and the comparison results between groups (with and without postural changes). Among the PA variables, a statistically significant difference was only observed in the Frequency of PA for the *pes planus* condition, since the individuals without postural change reported a higher score than the ones affected by postural change ($p$ = 0.035).

**Table 4.** Descriptive statistics for physical activity variables according to foot postural changes ($n$ = 231).

| Variables | Pes Planus | | | | Pes Cavus | | | |
|---|---|---|---|---|---|---|---|---|
| | **With** | **Without** | **Comparison** | | **With** | **Without** | **Comparison** | |
| | **Mean ± SD** | **Mean ± SD** | **t** | **p** | **Mean ± SD** | **Mean ± SD** | **t** | **p** |
| Score Formal PA | 11.32 ± 2.57 | 12.24 ± 3.10 | 1.869 | 0.063 | 11.44 ± 2.01 | 12.08 ± 3.03 | 0.560 | 0.576 |
| Score Informal PA | 2.82 ± 0.55 | 2.79 ± 0.56 | −0.316 | 0.752 | 2.79 ± 0.56 | 2.79 ± 0.38 | −0.685 | 0.494 |
| Score Total | 14.13 ± 2.86 | 15.05 ± 3.26 | 1.738 | 0.068 | 14.36 ± 2.22 | 14.82 ± 3.21 | 0.421 | 0.674 |
| Practical History | 7.42 ± 4.47 | 10.52 ± 5.69 | 3.393 | 0.001 * | 12.17 ± 5.98 | 9.52 ± 5.51 | −1.401 | 0.163 |
| Frequency of PA | 6.78 ± 3.12 | 8.69 ± 4.41 | 2.130 | 0.035 * | 8.64 ± 2.78 | 8.19 ± 4.29 | −0.272 | 0.786 |

PA (physical activity); * $p \leq 0.05$.

## 4. Discussion

This study aimed to evaluate the occurrence of knee and foot postural alterations and the differences in body composition and PA among young healthy adults. The frequency of knee alterations (*genu recurvatum* 54.1%, *knee valgus* 22.9%, and *knee varus* 8.7%) and foot alterations (*pes planus* 25.5% and *pes cavus* 4.3%) in the present study were substantial. Concerning postural alterations for gender, a distinction emerged. *Genu recurvatum* was more common among females (68.2%) compared to males (48.2%), while *knee valgus* exhibited a significantly higher occurrence in males (60.4%) than in females (39.6%).

Indeed, the literature has underlined a heightened occurrence of *genu recurvatum* among females compared to men [34–36]. In line with these findings, previous authors Penha, et al. [37] reported a comparable prevalence of *genu recurvatum* (54%) in school-aged children. Conversely, Gh, et al. [38] revealed that 22% of children exhibited *genu recurvatum* at birth, which showed slight variation with age or gender. The reason for the higher prevalence of *genu recurvatum* is not well known. However, repetitive and harmful habits that led to posterior capsule laxity could be associated [39], as well as a lack of strength/weakness of the gastrocnemius muscle [40] and the quadriceps muscle, which allows hyperextension of the knee [41]. In addition, gender differences may arise from factors such as greater knee laxity exhibited by females [42,43], knee geometry variations, and smaller anterior cruciate ligaments [44,45]. Further insights are drawn from another study highlighting the predominance of *genu varus* as a knee alteration, particularly pronounced among females [46]. In contrast, an alternative investigation by Odding, et al. [47] described *varus* deformities as more pronounced in males, while *valgus* deformities exhibited greater prevalence in females. A study conducted on the prevalence of dynamic knee valgus among children indicates 26.3% and 26.9% in the right and left lower limbs, with females exhibiting more *knee valgus* in the left limb [48]. Concerning foot alterations, prior research showcased the presence of *pes planus* in a staggering 90.8% of elementary school students [49]. However, among young, healthy individuals, *pes planus* was 29% in the South Indian population [50]. Notably, the occurrence of *pes cavus* (4.3%) was consistent with previous studies on the topic that reported a prevalence ranging between 0.2 to 3.7% [49,51–54].

Furthermore, in the current study, *genu recurvatum* condition was statistically significantly related to body composition parameters, particularly %FM and WHR. However, individuals without *genu recurvatum* showed statistically significant higher mean values of TBW, protein, minerals, and SMM. These findings align with the concept that diverse body types give rise to disparities in fat distribution, often manifesting as increased fat mass [55]. The potential connection between body composition and *genu recurvatum* can be attributed

to muscle strength imbalances stemming from these varying fat distributions, thus contributing to muscular weaknesses and imbalances [56,57]. In the case of *knee varus* and valgus, no statistically significant connection was established with body composition, given the particular locations and age group in our sample with normal BMI, excluding obesity as a confounding factor. Nevertheless, a study involving individuals aged 11 and 13 years established a significant relationship between weight and *knee varus* and stature and *knee varus* [58]. In addition, the literature presents varying perspectives on the nexus between *genu varus*, *genu valgum*, and obesity. While, Soheilipour, et al. [59] proposed a weaker connection between *genu varus* and obesity, the authors noted a heightened prevalence of *genu valgum* in obese individuals. This observation gains reinforcement from multiple studies collectively indicating an increased occurrence of *genu valgum* [59–62]. Moreover, in the present study, no statistically significant relationship was observed between *pes planus*, *pes cavus*, and BMI, except between *pes cavus* and stature. This contrasts with previous research that asserts a robust and highly significant connection between *pes planus* and BMI [49,63,64].

Concerning PA and postural alterations, no statistical difference was observed between groups with and without *genu recurvatum*; however, formal PA was significantly linked to *knee valgus* and *varus*. The total PA (formal and informal) also differed notably among *knee varus* conditions. Strenuous knee-involved activities can lead to muscle tightness and joint strain, potentially contributing to *knee varus*. This aligns with Lee [65], who emphasized that stress during PA, especially among overweight individuals aiming to lose weight, might precipitate *knee varus* before weight reduction. The reason for *knee valgus* and varus relation with PA is still unknown.

Nonetheless, prevailing research emphasizes how high-intensity athletic performance can induce global and regional muscle fatigue, impairing postural stability [66–68]. Moreover, training parameters affect general fatigue, leading to a strong relationship between the type of exercise, fatigue, and postural deficits [68,69]. Notably, varying postural stability levels were observed in physical education students after short, intense, or prolonged moderate exercise [69]. Evidence suggests that training with poor posture can deteriorate the muscles' proprioceptive feedback mechanism, limiting their ability to correct stance and maintain proper posture due to reduced sensory system input [67,70,71]. Muscle fatigue in stabilizing muscles (such as the gastrocnemius and soleus) misaligns joints and weakens neuromuscular control, contributing to decreased postural stability [66,72]. Furthermore, a statistically significant relationship is found between *pes planus* and practical history and physical exercise frequency. Possible causes include the laxity of soft tissues supporting the arch, including the tibialis posterior muscle, plantar fascia, intrinsic foot muscles, and calcaneonavicular ligaments [73]. A study conducted in Turkey reported weakness of foot plantar flexor group muscles, reduced flexibility of gastrocnemius and soleus muscles, and decreased balance with physical activity in individuals with *pes planus* [74]. In the current study, no statistical significance was found between groups with and without *pes cavus* and physical activity. The lower occurrence could be attributed to congenital causes or neuromuscular disorders like muscular dystrophy and Charcot–Marie–Tooth disease [75,76]. Traumatic injuries can also impact the tarsal bone position and lead to hypertonicity of the longitudinal arches. In *pes cavus*, the peroneus longus and posterior tibialis muscles tend to overpower the peroneus brevis, tibialis anterior, and intrinsic foot muscles, often resulting in plantar flexion due to peroneus longus contracture [77]. Studies have shown that foot postural alterations like *pes planus* and *pes cavus* are associated with an increased risk of various lower extremity injuries compared to individuals with a neutral arch [78–80].

Although these are important findings, this study presents some limitations that should be mentioned. Notably, the study's sample size is relatively modest, suggesting the potential for enhanced robustness by broadening the scope to encompass a more diverse and extensive population. Future efforts could emphasize recruitment across different demographic groups and regions to fortify the generalizability of findings. Moreover, this study's methodology relies on an observational analysis of postures, which, while

informative, represents only a partial assessment of the complexities involved. Future investigations could incorporate a multifaceted approach to enhance the depth of research. A further limitation lies in the absence of interventions to address observed postural alterations among the student participants. A valuable avenue for future research lies in implementing interventions tailored to correct identified postural discrepancies and measuring the resultant changes. This longitudinal approach would not only elucidate the potential efficacy of intervention strategies but also contribute to developing evidence-based recommendations for managing and improving posture-related concerns. This study presents a starting point to understand the occurrence of knee and foot postural alterations according to the individuals' body composition and PA profiles, which could support the deployment of tailored interventions among healthy adults.

## 5. Conclusions

Understanding the prevalence of knee and foot postural alterations holds substantial importance in preventing issues like shin splints, stress fractures, and plantar fasciitis during both physical exertion and routine activities. This study sought to pinpoint key determinants that play a pivotal role in discerning shifts in knee and foot posture while exploring their connections with body composition and physical activity. Ultimately, the investigation highlighted noteworthy factors such as BMI, WHR, and past PA experiences, among others, that could significantly contribute to devising effective strategies for addressing postural alterations in the knee and foot.

**Author Contributions:** Conceptualization, S.A., R.V., R.T.O. and A.R.; methodology, S.A., R.V., R.T.O. and A.R.; software, A.R.; validation, S.A., R.V., C.F., R.T.O. and A.R.; formal analysis, S.A., C.F. and A.R.; investigation, S.A. and R.V.; resources, S.A., R.V., C.F., R.T.O. and A.R.; writing—original draft preparation, S.A., C.F. and A.R.; writing—review and editing, S.A., R.V., C.F., R.T.O. and A.R.; visualization, C.F., R.T.O. and A.R.; supervision, R.T.O. and A.R.; project administration, R.T.O. and A.R.; funding acquisition, C.F., R.T.O. and A.R. All authors have read and agreed to the published version of the manuscript.

**Funding:** C.F. recognizes the support from LARSyS—the Portuguese national funding agency for science, research, and technology (FCT) pluriannual funding 2020–2023 (ITI-LX UIDP/50009/2020—IST-ID).

**Institutional Review Board Statement:** The study was conducted according to the guidelines of the Declaration of Helsinki and approved by the Scientific Committee of The Faculty of Physical Education and Sports at the University of Madeira (reference: ACTA N.77-12 April 2016).

**Informed Consent Statement:** Informed consent was obtained from all individuals involved in the study.

**Data Availability Statement:** The data presented in this study are available upon request from the corresponding author.

**Conflicts of Interest:** The authors declare no conflict of interest.

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
