# Peer review of "Association between Body Composition, Physical Activity Profile, and Occurrence of Knee and Foot Postural Alterations among Young Healthy Adults"

_future, doi:10.3390/future2010002_

Round 1

Reviewer 1 Report

Comments and Suggestions for Authors

The authors describe the relationship between knee and foot deformities correlated with young healthy adults' body composition and physical activity. The manuscript is well written. However, there are some issues.

Abstract: OK

Introduction: OK

Methods: -a flow chart is missing -describing the number of patients, inclusion-exclusion criteria  

                  - the power of study is missing.

                 -also, confidence intervals should be clearly stated

                  -

Results OK. Probably tables 2 and 3  should be \ presented as graphs.

Discussion: Authors must try to expand the discussion section by including and discussing more literature data.

Conclusions: OK

Reviewer 2 Report

Comments and Suggestions for Authors

1. The number of female participants is only 40% of male participants. Thus, comparing the male to the female group is not appropriate. Additional tests to prove that they are comparable are needed.

2. The Latin terms in italic should be explained clearly. 

3. It would be better if the authors also presented the figures of how the participants did the tasks in the experiment. 

Comments on the Quality of English Language

Minor editing is required to fix some typos and misspellings. 

Reviewer 3 Report

Comments and Suggestions for Authors

Overview

The authors conducted a descriptive study to assess lower limb deformities and whether they are associated with physical activity and body composition.  Young adults at a university were given a postural evaluation for categorized as not having or having knee (i.e., genu recurvatum, knee valgus and vargus) and feet (i.e., pes planus and pes cavus) structural abnormalities.  Subjects were also measured for body mass, stature, waist and hip ratio, and body composition (BIA) and surveyed for physical activity.  The authors reported percentages of subjects within the sample having deformities, the proportion of differences for males and females, and that those with genu recurvatum carried more body fat particularly as central/visceral fat.  For physical activity, surprisingly, those with knee varus had higher scores for formal (planned) physical activity. Otherwise statistically significant differences did not appear between those with and without the leg conditions. The authors provide potential explanations and encourage future work on interventions.

Major Concerns

 The mean age is almost 23 y with a large standard deviation, +4.9 y.  A rough 95% CI would be 12 to 32 y.  If this captures the true population value, only a small number might be preadolescents and adolescents.  Since the journal’s focus is on children and adolescents, this study doesn’t seem to hit the right population.  One suggestion is for the authors to compare the minors within this sample to the adults, if enough minors were represented.  Or they could work in age or category (youth vs adult) as a factor in the analyses.

The authors have entitled this a study of relationships, but their study design involves group comparisons and correlation/regression was not used as the title would lead the reader to believe.  I appreciate the authors taking a conservative approach, but I think they should describe the approach and results accurately but continue to be conservative (e.g., potential causes or likely impact) in their statements.

 In lines 252-279 the authors discuss the possibility that physical activity might induce some of the deformities that they assessed.  Did the authors investigate whether the subjects were doing regimented exercise or physical therapy (PT) specifically for the conditions? In lines 289-290 the authors state “A further limitation lies in the absence of interventions to address observed postural alterations among the student participants.” Were interventions absent or did the authors just not probe whether subjects were engaged in interventions? Lines 74-77 do not indicate whether there were any exclusion criteria. Lines 120-125 identify how formal activity was evaluated but specifically not whether PT was occurring for the leg conditions.

Specific Suggestions and Comments

 Title: “Relationship” is stated.  It invokes the idea of using correlation and regression.  Instead, the authors use t tests, which might be more in line with testing for the effects of deformities on the key outcome variables in this cross-sectional study.  Given the categories are binary (does/does not have a specific deformity) logistic regression would be needed.

 Line 76: Please provide the minimum and maximum age values

 Line 144: The authors list 0.05 (5%) as the criterion for statistical differences but Table 2 identifies differences in ICF and SMM for knee valgus that are >0.05.  Remove the asterisk but free free to discussion them as close to being different.  Between Table 2 and 3, the authors have performed 55 t tests.  By chance, we’d expect a few to show up statistically significant but actually be a type 1 error.

 Line 187: Among the footnotes, abbreviations for BMI, TBW and WHR are not provided.  BMI is universal; maybe TBW and WHR, too.  I’d suggest adding them for the novice to this topic. 

Lines 136-144: As noted above, I expected to see correlation to be used, given the title.  Should the title be changed to “Potential effects of foot and knee deformities on…”?

Lines 163-165 “Furthermore, WHR (M = 0.86 ± 0.07 cm) and FM (M = 20.8 ± 10.1%) in individuals with genu recurvatum alteration were significantly superior to WHR (M = 0.82 ± 0.04 cm) and FM (M = 14.9 ± 7.0%) in individuals without genu recurvatum alteration.”  “Superior” denotes judgment and is subjective.  I suggest using objective terms, like higher or larger, to show direction of the difference.   Likewise in line 168, line 192, and elsewhere in the manuscript.

Line 196 “…did not present this postural change.:”  Did the authors measure change?  Or do they mean “challenge” since this study is a one-time cross-sectional evaluation?

Lines 236-237, “The potential connection between body composition and genu recurvatum can be attributed to muscle strength imbalances stemming from these varying fat distributions, thus contributing to muscular weaknesses and imbalances.:” The authors need to provide a reference of support or make this a more speculative comment since strength was not measured in this study.

Line 254-255 “The total PA (formal and informal) was also notably associated with knee varus.” Since t tests were used for group comparisons, I think the statement should say “different,” not associated.

Line 296 “…their relationships with body composition and PA levels…” Again, since relationships (correlation) was not truly investigated, I suggest the authors use “potential impact on…” instead of “relationships with…”

Lines 299-301 “Understanding the prevalence of knee and foot postural alterations holds substantial importance in preventing issues like shin splints, stress fractures, and plantar fasciitis during both physical exertion and routine activities.”  I’m not sure how the authors arrive at this without assessing shin splints etc. From their data, it appears people with structural alterations are less physically active and may be at less risk for the listed injuries that are common to overuse.

Comments on the Quality of English Language

The English is okay.  In several places, though, the authors use words that are subjective or judgmental (e.g., superior) when they should use objective words that describe the direction of the different (e.g., larger or greater).

Round 2

Reviewer 1 Report

Comments and Suggestions for Authors

The authors have addressed all of the issues. 

Author Response

Thank You for your valuable suggestions.

Reviewer 3 Report

Comments and Suggestions for Authors

The title is improved for accuracy but the final phrase “…considering their body composition and physical activity profile” falls short of describing the comparisons that were made, and the impact the leg conditions might have an impact on body composition and PA.

The authors continue to use the terms relationship and association when in fact comparisons were done as tests of possible effects.  Examples:

·        Line 21 “…body composition was positively related to the occurrence of genu recurvatum:” No statistic for the relationship.  “Body composition differed with the presence of genu recurvatum” would be accurate based on the authors’ design and analyses.

·        Line 82 “… (2) assess the PA and body composition profiles between individuals…:” The authors could make this stronger by stating it as “…(2) compare PA and body composition profiles…” since that is what they did.

·        Lines 279-280, “Concerning PA and postural alterations, it was observed that genu recurvatum displayed no statistically significant relationship…:” No statistic for the relationship was reported.  The authors should state that “…no statistical difference was observed between groups for…

·        Line 303-304 “Conversely, no statistical significance was found between pes cavus and physical activity.” Do the authors mean no significant difference was found for physical activity based on presence of pes cavus?  And is this statement in reference to the prior citation or the present study? If that latter, please add “in the current study.”
